# Muffin-Type Bakery Product Based on Mexican Mesquite (*Prosopis* spp.) Flour: Texture Profile, Acceptability, and Physicochemical Properties

**DOI:** 10.3390/foods12193587

**Published:** 2023-09-27

**Authors:** María Elizabeth Alemán-Huerta, Brenda A. Castillo-Cázares, Julia Mariana Márquez-Reyes, Juan G. Báez-González, Isela Quintero-Zapata, Fátima Lizeth Gandarilla-Pacheco, Erick de Jesús de Luna-Santillana, Mayra Z. Treviño-Garza

**Affiliations:** 1Facultad de Ciencias Biológicas, Instituto de Biotecnología, Universidad Autónoma de Nuevo León (UANL), Av. Pedro de Alba S/N, Cd. Universitaria, San Nicolás de los Garza 66455, Mexico; maria.alemanhr@uanl.edu.mx (M.E.A.-H.); brenda.castilloczr@uanl.edu.mx (B.A.C.-C.); isela.quinterozp@uanl.edu.mx (I.Q.-Z.); fatima.gandarillap@uanl.edu.mx (F.L.G.-P.); 2Facultad de Agronomía, Universidad Autónoma de Nuevo León (UANL), Francisco I. Madero S/N, Ex Hacienda el Cañada, Escobedo 66050, Mexico; julia.marquezrys@uanl.edu.mx; 3Departamento de Alimentos, Facultad de Ciencias Biológicas, Universidad Autónoma de Nuevo León (UANL), Av. Pedro de Alba S/N, Cd. Universitaria, San Nicolás de los Garza 66455, Mexico; juan.baezgn@uanl.edu.mx; 4Centro de Biotecnología Genómica, Instituto Politécnico Nacional, Blvd del Maestro S/N, Narciso Mendoza, Reynosa 88710, Mexico; edeluna@ipn.mx

**Keywords:** mesquite flour, bakery product, texture profile analysis, sensory analysis, physicochemical properties

## Abstract

In this research, muffin-type bakery products were developed based on wheat flour (WF) and mesquite flour (MF) in the following proportions: WFMF 90:10, WFMF 75:25, and WFMF 50:50. The products were characterized based on various properties in which it was possible to observe that the water activity (aw) did not show a significant change with the increase in the concentration of MF. In addition, the increase in the concentration of MF modified the sensory properties (color, odor, flavor, texture, and acceptance), further decreasing the luminosity and increasing the values of the a* and b* coordinates. Moreover, in the texture profile analysis, it was found that the increase in the MF concentration increased hardness, fracturability, and gumminess and decreased adhesiveness and cohesiveness. All the previously mentioned changes were more evident in the WFMF50:50 and, to a lesser degree, in WFMF75:25. In general, in most evaluations realized, the WFMF90:10 treatment was the most similar to the control (without MF). However, WFMMF75:25 provided a higher protein and fiber content and a lower fat content. Finally, it is possible to use the flour obtained from the mesquite fruit to make bakery products since it is an important source of food due to the wide distribution of mesquite in the country.

## 1. Introduction

Food is an essential factor for the well-being of the human being. Likewise, the lifestyle of today’s society has presented changes in recent years since the quantity and quality of food, as well as feeding habits, are factors that directly influence health. Therefore, the consumer is not only looking for sensory-acceptable foods but also healthy products that contain quality nutritional components that provide a beneficial effect on health [1,2,3].

Bakery products are among the most consumed foods worldwide [4]. Bread, cookies, cakes, and other products are widely consumed as desserts or snacks; however, these types of foods have a reputation for containing low-quality ingredients (high levels of sugar and saturated fat), high energy density (calories), high glycemic response, and low satiety effect [5]. In addition, excessive consumption of these products can contribute to an increased risk of certain diseases, such as obesity, heart disease, and diabetes.

Moreover, the most commonly used cereal for making bread is wheat flour (WF; which contains gluten protein), but this type of product is restricted for people with conditions, such as celiac disease (gluten intolerance), Dühring’s disease (dermatitis herpetiformis), gluten ataxia, gluten sensitivity (non-celiac), wheat allergy, autism spectrum disorders, among others, so they can only consume gluten-free bakery products [6]. Recently, exhaustive efforts have been made in the research area to develop innovative bakery products enriched with additives or healthy ingredients that provide health benefits [5]. In this context, for the improvement in the formulations of bakery products, there is a growing interest in gluten-free flours, such as rice, soy, buckwheat, and starches of various botanical origins [6]. Additionally, quinoa, amaranth, oats, corn, sorghum, chickpea, and peas, among others, have also been used [7].

Mesquites (*Prosopis* spp.) are species of the legume family found mainly in semi-arid and arid areas of Mexico. Most of the species are native to the Americas (40), some to Asia (3), and Africa (1) [8,9,10,11]. In addition, mesquite has been consumed since ancient times in countries such as Mexico, Peru, Brazil, Chile, and Argentina, among others. Mesquite flour is the product obtained by grinding the whole ripe pods of the *Prosopis* tree [3,12,13]. Mesquite flour is described as a sweet product of brown color, with an odor resembling coffee, cocoa, coconut, molasses, caramel, cinnamon, and hazelnut [3,13,14]. Thus, because of these characteristics, this product has been used extensively in the food industry as a substitute for cocoa in various products (e.g., cakes, candies, ice cream, and drinks, among others) [1].

On the other hand, some authors have reported that mesquite flour is a good source of fiber, amino acids, proteins, and minerals (potassium, calcium, and iron), as well as bioactive compounds with antioxidant effects (phenolic compounds such as flavones, condensed tannins, and gallic acid), anti-inflammatory and hypertensive. In addition, it has a low content of carbohydrates and fats [3,5,8,10,12,15]. Therefore, the incorporation of this raw material in the preparation of bakery products can provide a potential alternative to improve the dietary contribution of some macro and micronutrients [12,14].

Finally, mesquite flour has been used in the production of cakes [1], functional spread as a nutritious snack for children [2], gluten-free bread [14,16], muffins [5], panettone-like bread [4,12,17,18], sponge cakes [3], wheat bread partially replaced with Peruvian mesquite [10,13], sourdough bread [7], and loaf bread [6,11], among other products. These investigations have focused on the improvement of properties, as well as on the study of physicochemical, sensory, and textural properties, among others. Thus, the objective of this research was to prepare muffin-type bakery products based on mesquite flour and characterize them by analyzing the texture profile, sensory acceptance, and physicochemical properties.

## 2. Materials and Methods

Commercial mesquite flour purchased from “Producto Oaxaca”, Oaxaca de Juárez, Oaxaca, México (Table 1) was used to prepare bakery products. Wheat-based composite flour (Bakery mix, product number 00722042, Dawn^MR^, Dawn Foods México, Naucalpan de Juárez, Mexico), vegetable oil (pure canola oil, ACH^®^ Foods Mexico, San Pedro Garza García, Mexico), chicken eggs, and purified water were purchased at a local supermarket.

### 2.1. Preparation of Bakery Products Based on Mesquite Flour

The products were made using the formulations shown in Table 2. The water, oil, and egg content was the same for the rest of the treatments. However, the wheat flour was partially replaced with mesquite flour in the following proportions: WFMF 90:10, WFMF 75:25, and WFMF 50:50. The compounds were placed in a metal container and homogenized in a stand mixer (one batch per treatment at 25 ± 2 °C) until a uniform mixture was obtained. The mixture (40 g) was placed in muffin molds (4 cm in diameter) and then cooked in a conventional oven (Whirlpool, WFR3100B, Celaya, Mexico) at 210 °C for 22 min. Subsequently, muffins were allowed to cool at room temperature for 15 min and later packaged in Ziploc plastic bags until further analysis.

### 2.2. Preliminary Preference Test

The acceptance test consisted of evaluating the attributes of color, odor, flavor, texture, and overall acceptance, according to the 5-point structured hedonic scale (Scale: 1—dislike extremely; 2—dislike slightly; 3—neither like nor dislike; 4—like slightly and 5—like extremely), where scores ≥2.5 were considered acceptable [19]. The panel of tasters consisted of 20 untrained judges who like to consume muffin-type bakery products and whose ages ranged from 21 to 77 years of age (60% of the participants were women, and 40% were men). Panelists evaluated the attributes mentioned above; the samples were presented to the evaluators (in clear plastic plates) through identification with random three-digit codes. Evaluations were carried out in an illuminated area, with separate spaces, and at room temperature. The panelists used purified water to rinse their mouths after each sample was tested.

### 2.3. Water Activity, Color Analysis, and Texture Profile Analysis of Bakery Products

#### 2.3.1. Water Activity (Aw)

The aw of the crumb from each baked product (*n* = 3) was determined with the AquaLab Series 4TE equipment (Decagon Device, Inc., Washington, DC, USA) at a temperature of 25 °C [10].

#### 2.3.2. Color Measurements

Color determinations (*n* = 3) were performed with a colorimeter (Hunter Lab ColorFlex EZ, Reston, VA, USA) according to the parameters of luminosity (L; 0 black to 100 white), a* coordinate (−green to +red), and b* coordinate (−blue to +yellow); measurements were made on the crust and in the crumb of the product [19,20]. Finally, ∆E (overall color difference) was calculated according to the following equation: ∆E* = [(L_c_ − L)^2^ + (a_c_ − a)^2^ + (b_c_ − b)^2^]^1/2^, where L_c_, a_c_, and b_c_ represent the color measurements of control samples, and L, a, and b are the color measurements of the different treatments [5].

#### 2.3.3. Texture Profile Analysis (TPA)

The TPA measurements of the products (*n* = 3) were performed in a Brookfield texture analyzer (CT3, Middleboro, MA, USA) using the TA11/1000 probe, with an activation load of 0.07 N at a test speed of 0.50 mm/s. The parameters evaluated were fracturability (N), hardness (N), cohesiveness (dimensionless), adhesiveness (J), springiness (dimensionless), gumminess (N), and chewiness (J).

#### 2.3.4. Bromatological Analysis

The bakery products with the best physicochemical and sensory properties (WFMF 90:10 and control) were selected for a bromatological analysis. The determinations made were moisture (dehydration in an oven at 105 °C; AOAC 950.46B) [21], ash content (combustion at 550 °C for 12 h; AOAC 938.08) [21]; crude protein (Kjeldahl method, Nx6.25; AOAC 928.08) [21], fat (ethereal extraction method; AOAC 933.05) [21], crude fiber (AOAC 962.09) [21]. Finally, total carbohydrates were calculated by difference from the other fractions of food (% Carbohydrates = 100 − [moisture (%) + ash (%) + proteins (%) + fat (%)]; Rosa et al. [1]).

### 2.4. Statistical Analysis

The results of the sensory, texture profile, and physicochemical analyses were subjected to an analysis of variance (ANOVA) and Tukey’s method with a confidence level of 95% (*p* ≤ 0.05), using the SPSS software (IBM version 22, SPSS Inc., Chicago, IL, USA).

## 3. Results and Discussion

### 3.1. Preliminary Preference Test of Bakery Products

In the case of color sensory evaluations, significant differences (*p* < 0.05) were found among the different treatments. The highest scores were for the control (4.50 ± 0.69), followed by WFWM 90:10 (4.35 ± 0.75), WFWM 75:25 (4.30 ± 0.92), and WFWM 50:50 (3.40 ± 1.05); this effect can be attributed to a color change in the products as the mesquite flour content in the formulations increases (Figure 1 and Figure 2a). These results agree with the data reported by Rosa et al. [1] and Korus et al. [6], who noted a darkening in the color of the product as the content of mesquite flour increased. However, all the products incorporated with mesquite flour presented scores within the color acceptance limit (≥2.5).

Regarding the odor analysis, the values of the control (4.55 ± 0.69), WFMF 90:10 (4.65 ± 0.67), and WFMF 75:25 (4.70 ± 0.47) were significantly similar. The WFMF 50:50 product presented the lowest scores (3.15 ± 0.69) but within the acceptance limit (≥2.5; Figure 2b). According to what was reported by Korus et al. [6], the incorporation of mesquite flour in the bread improved the smell of the product; this effect was related to the introduction of components that, during thermal processing, formed volatile derivatives that enriched the smell of bread. Moreover, Barba de la Rosa et al. [22] mention that the incorporation of mesquite flour can provide a strong odor in the product, as described by our sensory panelists in the WFMF 50:50 product.

In the flavor attribute, no significant difference was found (*p* > 0.05) among the scores of WFMF 75:25 (4.20 ± 0.95), WFMF 90:10 (4.30 ± 0.73), and control (4.50 ± 0.89), whose scores fluctuated between 4.2 and 4.5. Nonetheless, the lowest values (*p* < 0.05) were for WFMF 50:50 (3.15 ± 1.09) since the high concentration of mesquite flour modified the characteristic flavor of the muffin (Figure 2c). According to what was reported by Bigne (2018) [12] and Korus et al. [11], the incorporation of mesquite flour had a positive effect on the acceptability of flavor in concentrations of 15–25% and 7.5–10%, respectively. However, other studies report that the high tannin content in some flours has also been associated with a slightly bitter, acrid, and astringent aftertaste that may negatively affect consumer acceptance [1,2,16]. In addition, Afifi et al. [9] mention that *Prosopis* flour is generally used in concentrations of 10–25% since the flavor becomes too strong for most palates.

Regarding the sensory texture attribute, the values of WFMF 90:10, WFMF 75:25, and the control were significantly similar (*p* > 0.05), with scores that fluctuated between 4.45 and 4.65. Furthermore, the lowest scores (*p* < 0.05) were for WFMF 50:50 (3.60 ± 1.27; Figure 2d), indicating that the increase in the concentration of mesquite flour in the formulation decreases the texture attribute since the panelists reported a slightly harder product as compared to the other treatments, which showed a spongy and delicate texture; a similar behavior was reported in previous studies by Rosa et al. [1], Różyło et al. [16], and, Bigne et al. [12].

In general, although all the products were within the limit of sensory acceptance in all attributes, the highest overall acceptance score was for the control (4.55 ± 0.76), followed by WFMF 90:10 (4.30 ± 0.66), WFMF 75:25 (4.20 ± 0.89) and WFMF 50:50 (3.25 ± 1.02) (Figure 2e), indicating that mesquite flour is a good alternative for the development of bakery products with sensorially acceptable properties, as has been shown reported in previous studies [6,11,12].

### 3.2. Aw, Color Properties, and Texture Profile of Bakery Products

Regarding the aw determinations in bakery products, the values fluctuated between 0.89 and 0.91, below the values found in two studies carried out by Gonzales-Barron et al. [10,13] in wheat-mesquite flour bread (0.969–0.975), and wheat bread partially replaced with Peruvian mesquite (*Prosopis pallida*) flour (0.972–0.978); both studies with substitution levels of up to 15% flour. This effect could be related to higher protein content in the mesquite flour used in our research; it has been reported that some globular proteins can absorb large amounts of water, thus decreasing the water activity of breadcrumbs. Additionally, the presence of carbohydrates, such as galactomannan, retains moisture, forming a gel-like structure that directly influences the aw of the product.

In the color analysis, it was found that the incorporation of mesquite flour significantly modified (*p* < 0.05) the values of the L, a*, and b* coordinates in crumb and crust. In the case of luminosity, the values were higher for the control (76.15 and 63.66), followed by the WFWM 90:10 (60.26 and 51.45), and WFWM 75:25 (46.83 and 40.56) treatments; the lowest value was for WFWM 50:50 (41.22 and 32.29) (Figure 3a). This result is in agreement with that reported by Korus et al. [6,11], who reported a decrease in crumb lightness due to the incorporation of mesquite flour (5–10%) rich in proteinic substances and simple sugars in gluten-free dough and bread.

In the case of a*, a significant increase (*p* < 0.05) was observed in this parameter as the concentration of mesquite flour in the formulation increased (Figure 3b). The control showed the lowest values (2.41 and 7.80), followed by WFWM 90:10 (5.70 and 10.07), WFWM 75:25 (8.48 and 11.63), and WFWM 50:50 (8.58 and 9.19). These results agree with what was reported by Korus et al. [6,11], who report an increase in a* values as the mesquite flour content in the formulations increases (5–10%); this behavior could be related to a darker color of the mesquite flour and the possible formation of Maillard reaction products during baking. Pawłowska et al. [5] note that, although the products of the Maillard reaction were the main agents responsible for the formation of a reddish color in the muffins (crust and crumb), the cocoa powder incorporated into these products can be replaced by carob powder causing only small visible changes in the crumb; this behavior can be explained by a lower concentration of carob in the formulation (5%).

On the other hand, a significant decrease (*p* < 0.05) was found in the values of the b* coordinate as the amount of mesquite flour in the formulation increased (Figure 3c); the highest values were for the control (26.94 and 37.43) and WFWM 90:10 (28.13 and 32.35), followed by WFWM 75:25 (27.28 and 26.25) and WFWM 50:50 (23.61 and 16.61). A similar behavior was reported by Bigne et al. [12] in panettone-like bread, whose results show a significant decrease in the b* values as the mesquite flour content increases in the formulation. Furthermore, Różyło et al. [16] found that the b* value (yellowness) of the crumb decreased with increasing carob content in gluten-free bread enriched with carob fiber. Likewise, a similar effect was reported by Aydin and Özdemir [2] in a functional spreadable product based on carob flour, in which it was observed that due to the dark-reddish color of the carob flour, both the luminosity values decreased, as well as the greenness and yellowness of the product. In general, we can highlight that the highest values in color change (ΔE* in crumb and crust; Figure 4c) were for WFMF 50:50 (35.63 ± 0.38 and 37.87 ± 1.13), and the lowest values were for WFMF 90:10 (16.27 ± 1.00 and 14.06 ± 2.66). Finally, intermediate values were found in the WFMF 75:25 treatment (29.95 ± 1.59 and 26.27 ± 1.49) as observed in the determinations of L*, a* and b*.

Regarding the texture profile analysis, hardness refers to the force required to compress food between the molars or tongue and the palate [23,24]. The hardness of the products increased (*p* < 0.05) with the increase in the mesquite gum concentration, particularly in the WFMF 50:50 treatment, whose value was 10.47 ± 1.20 N (Figure 4a). On the other hand, the values of WFMF 75:25 were 5.37 ± 0.10 N, while the values of the control and WFMF 90:10 fluctuated between 2.66 and 3.01 N. Similar behavior was reported by Bigne et al. and Korus et al. [11,12], who found that the increase of mesquite flour in the formulations led to a progressive hardening of bread. According to Bing et al. [12], this behavior could be related to a higher fiber, protein, sugar, and mineral content in mesquite flour, which can affect the rheological and textural properties of the product.

The adhesiveness parameter represents the work required to overcome the attractive forces of the food and the surface of the materials with which the food comes into contact [23,24]. In the determinations of adhesiveness, a significant decrease (*p* < 0.05) was found when increasing the concentration of mesquite flour; the highest values were for the control (3.30 × 10^−4^) and WFMF 90:10 (3.00 × 10^−4^), while the lowest values were for WFMF 75:25 (7.00 × 10^−5^) and WFMF 50:50 (3.00 × 10^−5^) (Figure 4b). Compared to the results reported by Bigne et al. [12], the values found in our research are lower, in addition to the fact that these researchers found an increase in adhesiveness as the concentration of mesquite flour increased. According to what was reported by Gonzales-Barrón et al. [13], the properties of bread dough depend on the type of wheat flour used for its preparation since it has been found that wheat flour with a higher gluten content increases stickiness and work of adhesion. Thus, the decrease in adhesiveness in the WFMF 75:25 and WFMF 50:50 treatments could be related to a reduction in gluten content and the presence of mesquite flour components, such as galactomannans and fiber, which can modify the interaction between gluten proteins.

Regarding the fracturability parameter, a significant increase (*p* < 0.05) was observed as the concentration of mesquite gum in the formulation increased, the highest values being for WFMF 50:50 (10.47 ± 1.20 N), followed by WFMF 75:25 (5.37 ± 0.10 N), WFMF 90:10 (1.39 ± 0.78 N), and control (0.32 ± 0.16 N) (Figure 4c).

Fracturability or brittleness refers to the hardness with which the food crumbles or breaks [24,25]. Therefore, since fracturability is related to the hardness of the bread, we can point out that the greater the hardness of the crumb, the more brittle and fracturable it becomes, according to the findings in the evaluation of the hardness of the product.

The cohesiveness measurement indicates the ability to withstand a break under compression, and it represents the internal capacity of the sample to remain integrated [12,24]. In this parameter, the control, WFMF 90:10, and WFMF 75:25 formulations presented similar values (*p* > 0.05), which fluctuated between 0.67 and 0.71, while the lowest value was for the WFMF 50:50 treatment (0.62 ± 0.01; *p* < 0.05) (Figure 4d). According to what was reported by Rosa et al. [1], cohesiveness represents the internal structure of the product, which is associated with the molecular interactions of the bread components, where low values indicate a product that is difficult to handle and cut. This indicates that high concentrations of mesquite flour decrease the cohesiveness of the product, a behavior found to be related to higher fiber content in the product. Likewise, Bigne et al. [12] report that the cohesiveness usually diminishes when diluting gluten in composite breads, which indicates that the control treatments, WFMF 90:10 and WFMF 75:25 maintain a greater integrity of their food structure, while WF 50:50 is more susceptible to the disintegration [13].

This is with consideration of the elasticity parameter (a measure of how much of the original structure of the food has been broken down by the initial compression [23,24], although some studies report a reduction in the springiness of breadcrumbs associated with the increase in mesquite flour [12,13]. In our research, no significant difference (*p* < 0.05) was found between the different treatments, whose values fluctuated between 3.96 and 4.29 (Figure 4e), in agreement with what was reported by Korus et al. [6,11].

In the case of gumminess, this parameter simulates the energy required to break down a semisolid food so that it is ready to be swallowed, which is defined as the product of hardness times cohesiveness [23,24]. In our analyses, a significant increase (*p* < 0.05) in gumminess was found as the concentration of mesquite flour increased; the control presented the lowest value (1.88 ± 0.06 N), followed by WFMF 90:10 (2.08 ± 0.00 N) and WFMF 75:25 (3.60 ± 0.10 N). The highest value was found in the WFMF 50:50 treatment (6.52 ± 0.71 N) (Figure 4f). These results agree with what was reported by Bigne [17], who found that the gumminess values increase with the increase in the content of carob flour. As mentioned in the previous sections, this effect could be associated with the presence of mesquite flour components (such as galactomannans, proteins, and fiber) as well as the reduction in the gluten content, which can affect the textural properties of the product [12,13].

Concerning chewiness (an indicator for a mouthfeel that measures how much energy is required to chew a solid foodstuff before the food is suitable to swallow [23,24]), a significant increase (*p* < 0.05) was observed as the concentration of mesquite gum increased. Control and WFMF presented similar values (8.07 × 10^−3^–8.50 × 10^−3^ J; *p* > 0.05), followed by WFMF 75:25 (1.49 × 10^−2^) and WFMF 50:50 (2.59 × 10^−2^) (Figure 4g). This parameter reflects the measurement of the energy required to chew a solid and disintegrate it until it can be swallowed [24]. According to what was reported by Gonzales-Barron et al. [13], the chewiness of the bread crumb followed the same trend as the hardness as the content of mesquite flour in the formulations increased since these properties are related. This agrees with that found in our results of hardness and chewiness (Figure 4a,g). Likewise, Różyło et al. [16] found that the chewiness of the bread crumb significantly increased with the addition of carob fiber.

### 3.3. Bromatological Analysis

Concerning the moisture content and dry matter of the bakery products, the values found in the bakery products (28.86–32.03 and 71.15–67.94; Table 3) were similar to those reported by Gonzales-Barrón et al. [10], Goranova et al. [3], and González-Montemayor et al. [7] in bread products based on wheat flour and incorporated with mesquite flour.

Furthermore, regarding the mineral matter content, the values fluctuated between 1.75 and 2.69%; the lowest content was for the control, while the highest content was for WFMF 75:25 and WFMF 50:50 (Table 3). These values were higher than those reported by Torrelio-Martos et al. [14], Bigne et al. [12], Gonzales-Barron et al. [10] but lower than those reported by González-Montemayor et al. [7]. This behavior can be associated with the concentrations of mesquite flour used to produce the different bakery products, as well as the characteristics of the mesquite flour depending on the different regions of origin.

Regarding the protein content, some authors [1,12] do not report a significant increase in protein content between varying flour concentrations. However, in our study, we found that the incorporation of mesquite flour increased protein values from 6.07% (control) to 11.28% (WFMF 50:50) (Table 3). These findings are in agreement with what was reported by Gonzales-Barron et al. [10], who found an increase in protein content as the replacement level of mesquite flour increased in bread formulations.

Moreover, a decrease in the fat content was found when adding the mesquite flour in the formulations (up to 7.70% for WFMF 50:50 and 10.89% for the control samples; Table 3); these values are similar to those reported by Torrelio and López [14]. Additionally, these results are consistent with what was reported by Rosa et al. [1] and Gonzales-Barron et al. [13], who found a reduction in the fat content of the products due to the incorporation of mesquite flour; this effect could be associated with the fact that the fat content of the mesquite flour is low.

Concerning fiber content, a significant increase (*p* < 0.05) was observed in the treatments when adding mesquite flour in the formulations (0.57 for the control and up to 4.04% for WFMF 90:10; Table 3). These findings are in agreement with what was found in previous studies [7,10,12], in which it was reported that the fiber content increased with the addition of mesquite flour. Thus, this shows the beneficial effects that mesquite flour can provide for health due to the presence of polysaccharides such as galactomannans and lignin, which give it potential as a prebiotic for intestinal flora [14].

Finally, some studies do not show a significant effect on the carbohydrate content of increasing the amount of mesquite flour in the treatments [10]. However, our findings indicate that the incorporation of mesquite flour was able to reduce the carbohydrate content in baked goods (47.96% for WFMF90:10 and 51.80% for the control; Table 3), in agreement with what was reported by Rosa et al. [1]. In general, the values and variations in the content of protein, fat, fiber, and carbohydrates, among others, of the bakery products will depend on the characteristics of the mesquite flours used to develop the products.

## 4. Conclusions

In the preference test, it was evidenced that the increase in the concentration of mesquite flour modifies the organoleptic properties of color, smell, flavor, texture, and general acceptance, especially in the WFMF 50:50 treatment, which, although it was within the limits of acceptance, showed the lowest scores. The aw values did not show a significant effect when increasing the concentration of mesquite flour in the treatments. Moreover, in the color evaluations, a decrease in luminosity and the values of the b* coordinate was evidenced, as well as an increase in the values of the a* coordinate due to the incorporation of mesquite gum into the product, mainly in WFMF 50:50. Regarding the texture profile analysis, the increase in the mesquite gum concentration increased hardness, fracturability, and gumminess and decreased adhesiveness and cohesiveness, especially in the WFMF 75:25 and WFMF 50:50 treatments. On the other hand, although the WFMF 90:10 treatment showed few changes in most of the parameters evaluated with respect to the original product (control), in general, WFMF 75:25 provided a higher protein, minerals, and crude fiber content, as well as lower fat and carbohydrate content (nitrogen-free extract), which could be associated with beneficial health effects. Finally, mesquite is a plant species whose fruits can be used to make flour and incorporated into various foods such as bakery products. Moreover, at the same time, mesquite represents a source of healthy food for rural communities in much of Mexico due to the wide distribution of this species in the country.

## Figures and Tables

**Figure 1 foods-12-03587-f001:**
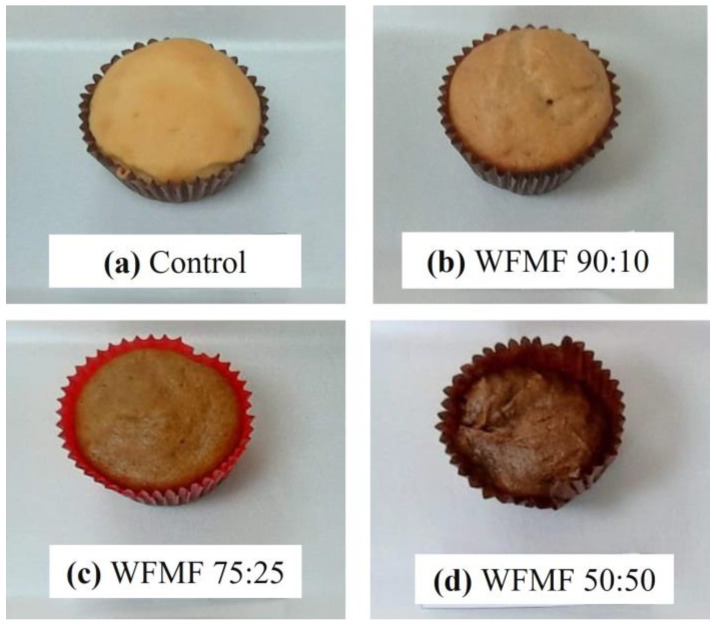
Representative image of bakery products made from mesquite flour in different proportions.

**Figure 2 foods-12-03587-f002:**
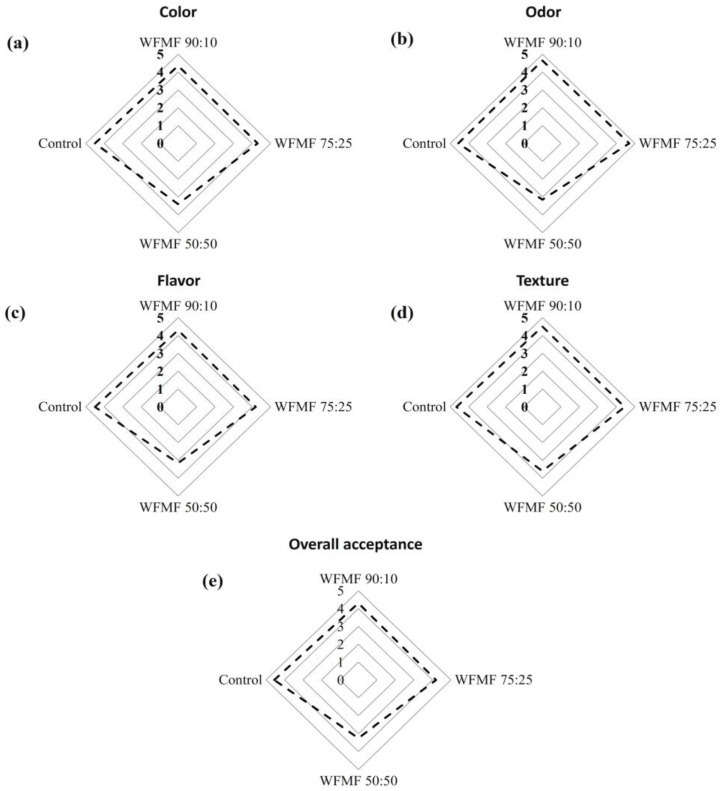
Sensory attributes of (**a**) color, (**b**) odor, (**c**) flavor, (**d**) texture, and (**e**) overall acceptance of mesquite flour-based bakery products in different proportions. The dotted line represents the average scores of the sensory tests.

**Figure 3 foods-12-03587-f003:**
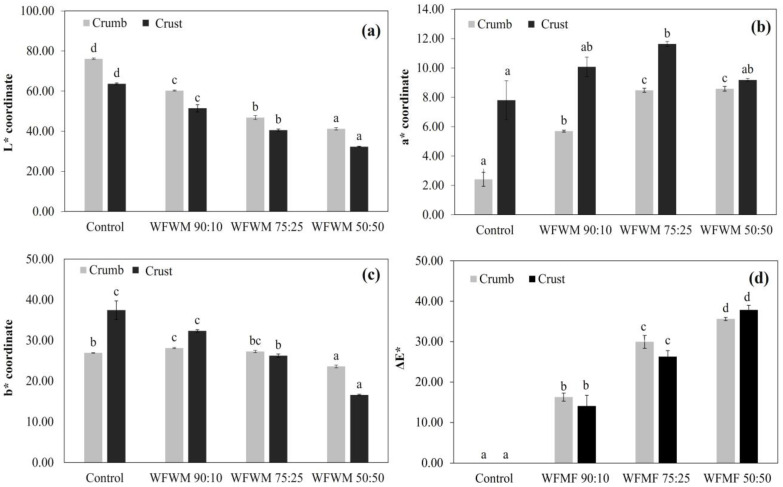
Color analysis: (**a**) Luminosity (black to white), (**b**) a* coordinate (−green to +red) and (**c**) b* coordinate (+yellow and −blue), and (**d**) ΔE* of mesquite flour-based bakery products in different proportions. Different letters (a–d) indicate significant differences (*p* < 0.05) between treatments.

**Figure 4 foods-12-03587-f004:**
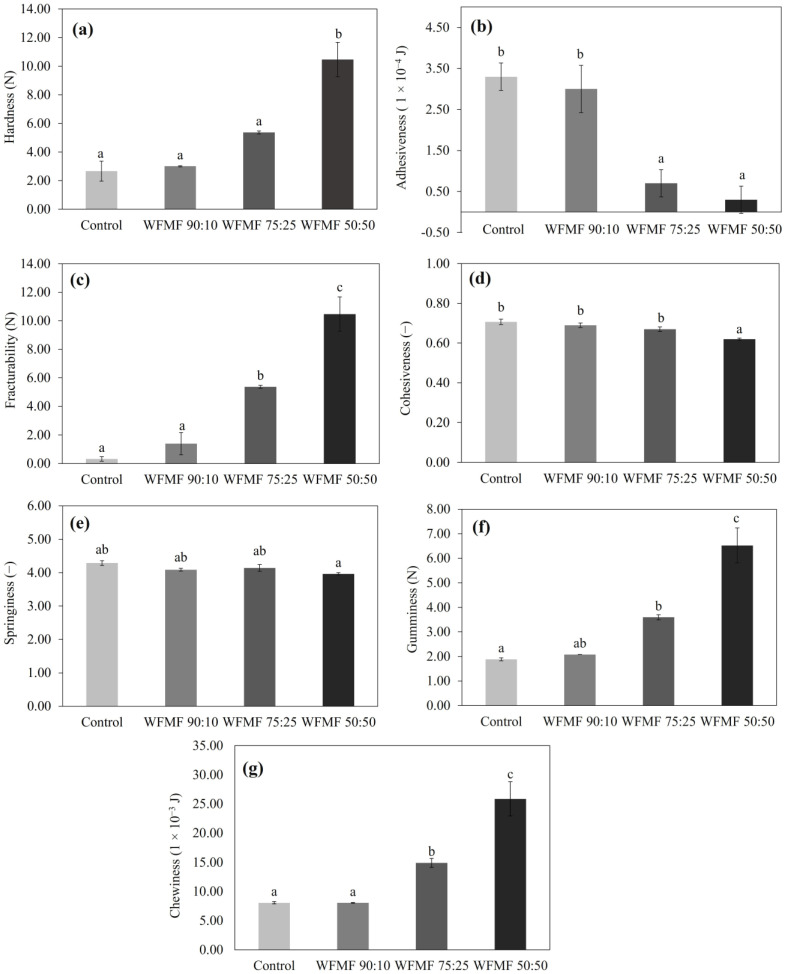
Texture profile analysis (TPA) (**a**) hardness, (**b**) adhesiveness, (**c**) fracturability, (**d**) cohesiveness, (**e**) springiness, (**f**) gumminess, and chewiness (**g**) of mesquite flour-based bakery products in different proportions. Different letters (a–c) indicate significant differences (*p* < 0.05) between treatments.

**Table 1 foods-12-03587-t001:** Bromatological composition of mesquite flour.

Composition	Value (%)
Moisture	5.83 ± 0.11
Dry matter	94.17 ± 0.11
Ash	4.61 ± 0.01
Protein	17.71 ± 0.12
Fat	9.57 ± 0.04
Crude fiber	7.88 ± 0.12
Carbohydrates	54.09 ± 0.16

**Table 2 foods-12-03587-t002:** Formulations to prepare bakery products.

Formulation	Water (%)	Oil (%)	Egg (%)	Wheat Based Flour (%)	Mesquite Flour (%)
Control	22.36	3.31	18.44	55.89	-
WFMF 90:10	22.36	3.31	18.44	50.30	5.59
WFMF 75:25	22.36	3.31	18.44	41.92	13.97
WFMF 50:50	22.36	3.31	18.44	27.95	27.95

**Table 3 foods-12-03587-t003:** Physicochemical analysis of mesquite flour-based bakery products in different proportions.

Physicochemical Analysis	Control	WFMF 90:10	WFMF 75:25	WFMF 50:50
Moisture	28.86 ± 0.00 ^b^	32.03 ± 0.01 ^d^	29.41 ± 0.08 ^c^	27.16 ± 0.10 ^a^
dry matter	71.15 ± 0.02 ^c^	67.94 ± 0.06 ^a^	70.59 ± 0.08 ^b^	72.84 ± 0.10 ^d^
Ash	1.83 ± 0.02 ^a^	1.75 ± 0.04 ^a^	2.24 ± 0.02 ^b^	2.69 ± 0.04 ^c^
Protein	6.07 ± 0.02 ^a^	6.33 ± 0.01 ^b^	8.86 ± 0.05 ^c^	11.28 ± 0.04 ^d^
Fat	10.89 ± 0.09 ^d^	9.90 ± 0.05 ^c^	8.45 ± 0.07 ^b^	7.70 ± 0.03 ^a^
Crude fiber	0.57 ± 0.00 ^a^	2.04 ± 0.01 ^b^	2.28 ± 0.06 ^c^	4.04 ± 0.13 ^d^
Carbohydrates	51.80 ± 0.11 ^b^	47.96 ± 0.01 ^a^	48.75 ± 0.11 ^a^	47.13 ± 0.08 ^a^

Note: Superscript letters (a–d) indicate significant differences between treatments.

## Data Availability

Data is contained within the article.

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
