# Peer review of "Muffin-Type Bakery Product Based on Mexican Mesquite (Prosopis spp.) Flour: Texture Profile, Acceptability, and Physicochemical Properties"

_foods, 2023, doi:10.3390/foods12193587_

Round 1

Reviewer 1 Report

After reading the manuscript "Muffin-type bakeryproduct based on mexican mesquite (Prosopis spp.) flour: texture profile analysis, acceptability, and physicochemical properties", I realized that the manuscript showed in some parts the scientific rigour wanted, but in other parts I have missed it.

The authors have presented critical evaluation only in some paragraphs.

The references are not exactly current, besides the objective/ title could be improved.

Thats why I have written some suggestions below in an attempt to improve the paper.

L.3- " analysis ...  properties" - review your title, please- Aren't they similar ? Do you really need to repeat ?

L.41- Be careful with "today", the references are from years ago.

L.44- "The"  Is it really necessary ?

L.56- Currently? Check the reference.

L.68- It seems important to me to mention which countries in the world consume this vegetable, you only presented production.

L.85- "chemical, bromatological" - Aren't they similar ? Do you really need to repeat ?

L.86- "rheological, and textural properties" -  Aren't they similar ? Do you really need to repeat ?

L.94- provide more details, for example which vegetable oil? Water - distilled ? temperature ? It seems a bit repetitive to have written and in the table 2, by the way I prefer the table, but I think it could be more detailed eith all the tretments. Please, include in table 2 mesquite flour and the formulations %.

L.72; L. 93; L.100- It became very repetitive "on the other hand"

L.96- I think it would be interesting to add the image of mesquite and its flour, many people think it is carob.

L.102- How many batches? Which oven was used? conventional? electric? combi? A unique oven was used for all the muffin repetitions.

L.112- For a sensory test a lot of relevant information was not included, I suggest reading papers and improving your article. Affective method ?  Acceptance test ?

Has the project been submitted to an evaluation by a university ethics committee? Did it follow the Helsinki declaration? Please, enter the approval protocol number. Which sensory test was performed? Were the analyses performed in sensory booths ? Did the assessors receive water to rinse the taste buds ? Are the assessors usually consumers of this product  ? 20 untrained ? Would you have authors to support this low number of assessors? Couldn't it have had a negative impact on your statistical analysis? Guys, the sensory part needs to be improved, a lot of important things are missing. It can't remain like this.   L131- Generally we call: crust and crumb. The same happened in Figure 4.   L.134- Only 3 repetitions? Isn't that too little for TPA?   L.151- Generally, authors follow the order of chemical, technological  and sensory evaluation, and in your paper it was sensory first. Just a comment. Think about ... and whether it makes any sense for the group.   L.154- between or among ?   L.188- I wonder if that's the best way to compare these results, it seems to me that it's not. The attributes title would be better above the spider graph.    L.198- I noticed that you used papers for discussion with carob, didn't you find any with mesquite?   de Gusmão, R. P., Cavalcanti-Mata, M. E. R. M., Duarte, M. E. M., & Gusmão, T. A. S. (2016). Particle size, morphological, rheological, physicochemical characterization and designation of minerals in mesquite flour (Proposis julifrora). Journal of Cereal Science69, 119-124.   de Melo Cavalcante, A. M., de Melo, A. M., da Silva, A. V. F., da Silva Neto, G. J., Barbi, R. C. T., Ikeda, M., ... & da Silva, O. S. (2022). Mesquite (Prosopis juliflora) grain flour: New ingredient with bioactive, nutritional and physical-chemical properties for food applications. Future Foods5, 100114.   L.240- Since the treatments are different, it would be interesting to use  different colors not only black (white, shades of gray) or hatchings. Think about this suggestion. I think you agree, i have just seen figures 5.   L.249 - "Figure x "  ??   L.305- I wonder if the results of Figure 5 wouldn't be better presented in a table, including their respective SDs.   L.386- You don't need to repeat % in every row, just once in the Physicochemical analysis cell    L.387-  between or among ?   L.389 - The conclusion you started with bromatological analysis, please reorganize it in the same sequence in all the sections of the paper. It gets confusing for readers this way.   L.396; L 402-It became very repetitive "on the other hand"  

Moderate editing of English language required

Author Response

After reading the manuscript "Muffin-type bakeryproduct based on mexican mesquite (Prosopis spp.) flour: texture profile analysis, acceptability, and physicochemical properties", I realized that the manuscript showed in some parts the scientific rigour wanted, but in other parts I have missed it.

The authors have presented critical evaluation only in some paragraphs.

The references are not exactly current, besides the objective/ title could be improved.

Thats why I have written some suggestions below in an attempt to improve the paper.

We appreciate the comments and suggestions of reviewer 1. Also, we clarify that this research included recent studies of the area such as González-Barrón et al. 2020, González-Montemayor et al. 2021, Korus et al. (2022), Zhong et al. (2022), among others.

L.3- " analysis ...  properties" - review your title, please- Aren't they similar? Do you really need to repeat ?

We agree, the title was modified.

L.41- Be careful with "today", the references are from years ago.

We agree, it was corrected.

L.44- "The" Is it really necessary?

We agree, it was corrected.

L.56- Currently? Check the reference.

We agree, it was corrected.

L.68- It seems important to me to mention which countries in the world consume this vegetable, you only presented production.

Ok. It was added.

L.85- "chemical, bromatological" - Aren't they similar ? Do you really need to repeat ?

We agree, it was corrected.

L.86- "rheological, and textural properties" -  Aren't they similar ? Do you really need to repeat ?

We agree, it was corrected.

L.94- provide more details, for example which vegetable oil? Water - distilled ? temperature ? It seems a bit repetitive to have written and in the table 2, by the way I prefer the table, but I think it could be more detailed eith all the tretments. Please, include in table 2 mesquite flour and the formulations %.

We agree, it was modified.

L.72; L. 93; L.100- It became very repetitive "on the other hand"

We agree, it was modified.

L.96- I think it would be interesting to add the image of mesquite and its flour, many people think it is carob.

We agree. Images of mesquite trees, whole pods and flour were included in graphical abstract.

L.102- How many batches? Which oven was used? conventional? electric? combi? A unique oven was used for all the muffin repetitions.

Ok. More information was included in the text.

L.112- For a sensory test a lot of relevant information was not included, I suggest reading papers and improving your article. Affective method ?  Acceptance test ?

Has the project been submitted to an evaluation by a university ethics committee? Did it follow the Helsinki declaration? Please, enter the approval protocol number. Which sensory test was performed? Were the analyses performed in sensory booths? Did the assessors receive water to rinse the taste buds? Are the assessors usually consumers of this product? 20 untrained? Would you have authors to support this low number of assessors? Couldn't it have had a negative impact on your statistical analysis? Guys, the sensory part needs to be improved, a lot of important things are missing. It can't remain like this.   

Ok. More information about acceptance tests was included in section 2.2 considering what was reported by previous authors (Barba de la Rosa et al., 2006; Różyło et al., 2017; Aydin et al., 2017; Bigne et al., 2018; Korus et al., 2022). Additionally, as reviewer 2 suggested, we change "sensory analysis" to "Preliminary preference test" since, as he tells us, an untrained panel may perform a preliminary evaluation of innovative products before consumer testing. Finally, since all the ingredients used are food grade, innocuous, safe, and widely commercialized, we consider informed consent for the panelists, where was explained the purpose of the research. Likewise, this project was approved by the academic committee of the institution following the code of ethics of the Faculty of Biological Sciences, UANL.

L131- Generally we call: crust and crumb. The same happened in Figure 4. 

We agree, it was modified.  

L.134- Only 3 repetitions? Isn't that too little for TPA?   

We considered the number of replicates for the TPA analysis based on previous studies (Korus et al., 2022).

L.151- Generally, authors follow the order of chemical, technological and sensory evaluation, and in your paper it was sensory first. Just a comment. Think about ... and whether it makes any sense for the group.  

We appreciate the comment. However, we consider that the order of appearance of the results does not affect their interpretation, so we would like to show them as they found in their current form.

L.154- between or among ?  

We agree, it was modified.  

L.188- I wonder if that's the best way to compare these results, it seems to me that it's not. The attributes title would be better above the spider graph.   

We agree, it was modified.  

L.198- I noticed that you used papers for discussion with carob, didn't you find any with mesquite? de Gusmão, R. P., Cavalcanti-Mata, M. E. R. M., Duarte, M. E. M., & Gusmão, T. A. S. (2016). Particle size, morphological, rheological, physicochemical characterization and designation of minerals in mesquite flour (Proposis julifrora). Journal of Cereal Science69, 119-124.   de Melo Cavalcante, A. M., de Melo, A. M., da Silva, A. V. F., da Silva Neto, G. J., Barbi, R. C. T., Ikeda, M., ... & da Silva, O. S. (2022). Mesquite (Prosopis juliflora) grain flour: New ingredient with bioactive, nutritional and physical-chemical properties for food applications. Future Foods5, 100114. 

We appreciate the comment. We included in this section the comparison of our results against studies of bakery products based on mesquite flour (Bigne et al., 2018) and carob flour (Rosa et al., 2015 and Różyło et al., 2017). We consider bakery products based on carob because it is a species of the genus Prosopis. Also, citations recommended by reviewer 1 (de Melo Cavalcante et al., 2022 and de Gusmão et al., 2016) were considered. However, these references were not included in the discussion section since these studies concern the characterization of mesquite flour but not the characterization of bakery products made with mesquite flour.

L.240- Since the treatments are different, it would be interesting to use different colors not only black (white, shades of gray) or hatchings. Think about this suggestion. I think you agree, i have just seen figures 5.   

We appreciate the comment. However, it is a matter of style. We consider that in the current state, the treatments can be adequately differentiated.

L.249 - "Figure x" ??   

There was an error. It was corrected.

L.305- I wonder if the results of Figure 5 wouldn't be better presented in a table, including their respective SDs.   

We appreciate the suggestion. However, we have found some studies where the TPA analysis is reported in graphs (including ± SD). Thus, we would like to represent the results of the TPA analysis as shown in their current state, since the point values of the different parameters are also mentioned in the wording of the results and discussion (section 3.2).

L.386- You don't need to repeat % in every row, just once in the Physicochemical analysis cell 

We agree, it was modified.  

L.387-  between or among ?  

…among others… it was revised.

L.389 - The conclusion you started with bromatological analysis, please reorganize it in the same sequence in all the sections of the paper. It gets confusing for readers this way.  

We agree, it was modified.

L.396; L 402-It became very repetitive "on the other hand"  

We agree, it was modified.

Reviewer 2 Report

The objective of submitted research was to prepare muffin-type bakery products based on mesquite flour, characterize them by analyzing the texture profile, and evaluate their sensory and physicochemical
properties. The title of the study fits well with its content. The subject of the study has novelty due to my literature review. This kind of muffin product has an industrial use potential. The introduction part well describes the need for mesquite flour into muffin and objective of the study, however it can be expanded with more novel literature. Materials and methods used in this study was well described, however it should include the necessary details regarding materials. More Statistical tools could be employed for optimization and to successfully emphasizing the differentiation among muffin samples, it is just a suggestion. In general, the obtained results were discussed well by referring to the related studies in the literature. The article is interesting and correctly written, however some issues need to be addressed by the authors.

Page 3, lines 99-109, specify the kind of oil and wheat based flour used for the muffin preparation;
Page 4, line 127, authors should calculate the values of total value color difference (∆E) in relation to the control sample;

From all figures letters marking order of figures should be deleted from and placed below the figures;

Page 8, line 249, please change Figure x to a proper numbering;

Page 10, move Figure 5 below the sentence were is mentioned for the first time;

Page 13, move Table 3 below the sentence were is mentioned for the first time;

Page 13, the Conclusion section must be rewritten, display conclusion in the same order as results are given in the Results and Discussions section

Author Response

The objective of submitted research was to prepare muffin-type bakery products based on mesquite flour, characterize them by analyzing the texture profile, and evaluate their sensory and physicochemical

properties. The title of the study fits well with its content. The subject of the study has novelty due to my literature review. This kind of muffin product has an industrial use potential. The introduction part well describes the need for mesquite flour into muffin and objective of the study, however it can be expanded with more novel literature. Materials and methods used in this study was well described, however it should include the necessary details regarding materials. More Statistical tools could be employed for optimization and to successfully emphasizing the differentiation among muffin samples, it is just a suggestion. In general, the obtained results were discussed well by referring to the related studies in the literature. The article is interesting and correctly written, however some issues need to be addressed by the authors.

We appreciate the comments and suggestions of reviewer 2.

Page 3, lines 99-109, specify the kind of oil and wheat based flour used for the muffin preparation;

Ok. The information was included in the text.

Page 4, line 127, authors should calculate the values of total value color difference (∆E) in relation to the control sample;

We agree. ∆E values were calculated and included in Figure 3. 

From all figures letters marking order of figures should be deleted from and placed below the figures;

We appreciate the suggestion. However, for a better understanding, we would like to keep the letters that mark the order of the figures, especially in large figures, such as Figures 2 and 4.

Page 8, line 249, please change Figure x to a proper numbering;

We agree, it was modified.

Page 10, move Figure 5 below the sentence were is mentioned for the first time;

We appreciate the suggestion. Figure 5 was modified. We are considering the arrangement of the figure (texture profile analysis), trying to optimize the spaces, and avoiding leaving as few spaces as possible on the page.

Page 13, move Table 3 below the sentence were is mentioned for the first time;

Ok. It was modified.

Page 13, the Conclusion section must be rewritten, display conclusion in the same order as results are given in the Results and Discussions section

We agree, it was modified. Conclusions are in the same order as results are given.

Reviewer 3 Report

Dear Authors

the paper propose the use of the mesquite flour to produce bakery products.

The subject is interesting, also because this plant is common in Mexico. Moreover, mesquite has some positive quality for human health and it is similar to cocoa.

I suggest you some modifications:

1-Please, check the size and the type of the font used in the Introduction is different from that used in the other part of the paper (Materials and Methods, Results and discussion, Conclusions).

2- Conclusion is chapter 4 not 5.

2-For a better understanding the results shown in Fig. 2 and 3 should be put toghether and the figures changed in only one figure. It is better to show the profiles of the 4 essays in one figure with all the parameters evaluated by the panel:  color,  odor, flavor, texture and overall acceptance.  

The paragraph lines 167-205 in which you described the results should be modified.

3- Chapter 2.2. and 3.1 Sensory Evaluations of bakery product

You wrote "Sensory evaluation" but it you should wrote "Preliminary acceptance evaluation". or "Preliminary preference test".

The panel did not describe the products or compare the products (for example with a ranking test).

Preference test should be conducted with a large number of consumers (miminum of about 50 people) but it is possible to have a preliminary evaluation of the innovative products done by a trained panel before the consumer test.

4-Please, change the term "sensory analysis" in all the text and use "preference test" or "acceptance test" for example line 391.

Revise the text for some minor revisions: sometimes the sentences are too long.

line 28: put a point instead of "semicolon"

Line 175-179: too long sentence, modify this paragraph.

Line 177: verify and check the meaning of: ,,,(3.15 ± 1.09; scor9; scores within the limit of sensory acceptance), ...

Line 193-199 too long sentence, modify this paragraph.

Line 316: after..flour [12,13], put a point instead of a semicolon after  [12,13], 

Line 339: after ..are related; this ....put a point instead of a semicolon after realted.

Line 359: after ... flour concentrations, put a point,  instead of a colon, c

Line 374: after "of mesquite flour," put a point,  instead of a colon, change the sentence if necessary.

line 380: after "treatments," put a point instead of a colon, change the sentence if necessary.

Line 406-410: too long sentence, divide the sentence, maybe after ..bakery products,  put a point. 

Author Response

Dear Authors

the paper propose the use of the mesquite flour to produce bakery products.

The subject is interesting, also because this plant is common in Mexico. Moreover, mesquite has some positive quality for human health and it is similar to cocoa.

I suggest you some modifications:

We appreciate the comments and suggestions of reviewer 1.

1-Please, check the size and the type of the font used in the Introduction is different from that used in the other part of the paper (Materials and Methods, Results and discussion, Conclusions).

Ok. It was modified.

2- Conclusion is chapter 4 not 5.

We agree, it was modified.

2-For a better understanding the results shown in Fig. 2 and 3 should be put toghether and the figures changed in only one figure. It is better to show the profiles of the 4 essays in one figure with all the parameters evaluated by the panel: color, odor, flavor, texture and overall acceptance.  

The paragraph lines 167-205 in which you described the results should be modified.

Ok. For a better understanding, some modifications were made in the wording.

3- Chapter 2.2. and 3.1 Sensory Evaluations of bakery product. You wrote "Sensory evaluation" but it you should wrote "Preliminary acceptance evaluation". or "Preliminary preference test". The panel did not describe the products or compare the products (for example with a ranking test). Preference test should be conducted with a large number of consumers (miminum of about 50 people) but it is possible to have a preliminary evaluation of the innovative products done by a trained panel before the consumer test.

We agree, it was modified.

4-Please, change the term "sensory analysis" in all the text and use "preference test" or "acceptance test" for example line 391.

We agree, it was modified.

Comments on the Quality of English Language

Revise the text for some minor revisions: sometimes the sentences are too long.

line 28: put a point instead of "semicolon"

We agree, it was modified.

Line 175-179: too long sentence, modify this paragraph.

Ok. It was modified.

Line 177: verify and check the meaning of: ,,,(3.15 ± 1.09; scor9; scores within the limit of sensory acceptance), ...

The limit of sensory acceptance (scores ≥2.5) was specified in section 2.2. Preliminary preference test (material and methods). Also, the paragraph was rewritten.

Line 193-199 too long sentence, modify this paragraph.

Ok. It was modified.

Line 316: after..flour [12,13], put a point instead of a semicolon after  [12,13], 

We agree, it was modified.

Line 339: after ..are related; this ....put a point instead of a semicolon after realted.

We agree, it was modified.

Line 359: after ... flour concentrations, put a point, instead of a colon, c

We agree, it was modified.

Line 374: after "of mesquite flour," put a point, instead of a colon, change the sentence if necessary.

We agree, it was modified.

line 380: after "treatments," put a point instead of a colon, change the sentence if necessary.

We agree, it was modified.

Line 406-410: too long sentence, divide the sentence, maybe after ..bakery products,  put a point. 

Ok. It was modified.

Round 2

Reviewer 1 Report

After another evaluation of the manuscript, I realized  improvement in the quality of the paper. The authors have accepted some of my requests.

They also have improved English, which is always useful to ask a native speaker for a final appreciation.

They added more authors to better substantiate the methodology and corrected tables and graphs. 

Some more suggestions below:

As you mention that the project has been submitted and approved by the Committee of the Faculty of Biological Sciences, UANL. Please, include the protocol number. This adds credibility to your paper.

Right. I agree with mentioning it as preliminary, possibly also due to the low number of evaluators who were untrained (n=20).

L.118 - between or among ?

Minor editing of English language required

Author Response

We appreciate the comments and suggestions of the reviewers regarding the manuscript entitled “Muffin-type bakery product based on Mexican mesquite (Prosopis spp.) flour: texture profile, acceptability, and physicochemical properties”. Additionally, we clarify that before submission the manuscript was subjected to an English language review.

After another evaluation of the manuscript, I realized  improvement in the quality of the paper. The authors have accepted some of my requests. We appreciate the reviewer's comment.

They also have improved English, which is always useful to ask a native speaker for a final appreciation.

Ok. We clarify that before submission, the manuscript was subjected to an English language review by a native speaker.

They added more authors to better substantiate the methodology and corrected tables and graphs. We appreciate the reviewer's comment.

Some more suggestions below:

As you mention that the project has been submitted and approved by the Committee of the Faculty of Biological Sciences, UANL. Please, include the protocol number. This adds credibility to your paper.

As reviewer 1 was informed, the project was approved by an academic committee of the institution since it was part of a thesis project, we do not have a protocol number as such. As previously reported, since all the ingredients used are food grade, innocuous, safe, and widely commercialized in our country, we consider only informed consent for the panelists (adults >18 years of age), where informed the purpose of the research, the importance of conducting sensory analysis, as well as the pros and cons of product consumption. Also, mesquite flour is an additive that has been used in the food area. In addition, there is no scientific information indicating that its consumption can affect health. Although this study did not undergo approval by an ethics committee, it was carried out following the ethical standards of our institution. We consider clarifying this information in the writing as shown below. We have reviewed and found this type of clarification in other scientific articles. We will consider this suggestion for future research.

Institutional Review Board Statement: This study was conducted according to the code of ethics of our institution, and it was not submitted to an ethics committee for validation (lines 427 – 428).

Informed Consent Statement: Informed consent was obtained from all subjects involved in the study (lines 430 – 431).

Right. I agree with mentioning it as preliminary, possibly also due to the low number of evaluators who were untrained (n=20).

We appreciate the reviewer's comment.

L.118 - between or among? Ok. It was modified.

Sincerely

Authors; María Elizabeth Alemán-Huerta, Brenda A. Castillo-Cázares, Julia Mariana Márquez-Reyes, Juan G. Báez-González, Isela Quintero-Zapata, Fátima Lizeth Gandarilla-Pacheco, Erick de Jesús de Luna-Santillana, Mayra Z. Treviño-Garza*
